# Comparison of the Surface Thermal Patterns of Horses and Donkeys in Infrared Thermography Images

**DOI:** 10.3390/ani10122201

**Published:** 2020-11-24

**Authors:** Małgorzata Domino, Michał Romaszewski, Tomasz Jasiński, Małgorzata Maśko

**Affiliations:** 1Veterinary Research Centre and Center for Biomedical Research, Department of Large Animal Diseases and Clinic, Institute of Veterinary Medicine, Warsaw University of Life Sciences (WULS–SGGW), 02-787 Warsaw, Poland; tomasz_jasinski@sggw.edu.pl; 2Institute of Theoretical and Applied Informatics, Polish Academy of Sciences, 44-100 Gliwice, Poland; mromaszewski@iitis.pl; 3Department of Animal Breeding, Institute of Animal Science, Warsaw University of Life Sciences (WULS–SGGW), 02-787 Warsaw, Poland; malgorzata_masko@sggw.edu.pl

**Keywords:** infrared thermography, equids, thermal patterns, surface temperature, skin thickness, hair coat

## Abstract

**Simple Summary:**

In this study, the thermal patterns of horses and donkeys in infrared thermography (IRT) images are analyzed and compared. Thermal patterns are defined as statistically significant differences between groups of regions of interest (ROIs) corresponding to underlying large muscles. The dataset used in the experiments consists of images of healthy and rested animals: sixteen horses and eighteen donkeys. Thermal patterns between species are compared, and the results are discussed along with special cases of animals identified as outliers. The results support the thesis about the similarities in the thermal patterns of horses and donkeys.

**Abstract:**

Infrared thermography (IRT) is a valuable diagnostic tool in equine veterinary medicine; however, little is known about its application to donkeys. This study aims to find patterns in thermal images of donkeys and horses and determine if these patterns share similarities. The study is carried out on 18 donkeys and 16 horses. All equids undergo thermal imaging with an infrared camera and measurement of the skin thickness and hair coat length. On the class maps of each thermal image, fifteen regions of interest (ROIs) are annotated and then combined into 10 groups of ROIs (GORs). The existence of statistically significant differences between surface temperatures in GORs is tested both “globally” for all animals of a given species and “locally” for each animal. Two special cases of animals that differed from the rest are also discussed. The results indicate that the majority of thermal patterns are similar for both species; however, average surface temperatures in horses (22.72±2.46 °C) are higher than in donkeys (18.88±2.30 °C). This could be related to differences in the skin thickness and hair coat. The patterns of both species are associated with GORs, rather than with an individual ROI, and there is a higher uniformity in the donkeys’ patterns.

## 1. Introduction

Infrared thermography (IRT) is a non-invasive imaging technique that allows for the detection of radiant energy emitted by any object with a temperature above absolute zero. The radiated power detected by the thermal camera in the infrared spectrum is proportional to the fourth power of the object’s absolute temperature, and it is used to calculate the temperature of the target, e.g., the surface of the animal’s body. Infrared radiation is often presented as a thermogram, which is an image where the color gradient corresponds to the distribution of surface temperatures [1]. Furthermore, the relationship of temperature gradients may create specific thermal patterns, which may be used, e.g., for assessing the influence of load on saddle fit in horses [2] or the horses’ response to the training [3].

IRT has been used as a diagnostic tool in equine veterinary medicine since the mid-1960s, particularly in the field of orthopedics, in the management of lameness [4,5,6,7]. The surface temperature changes, reflecting heat emitted from overloaded or injured tissue, are considered a valuable indicator for identifying areas of inflammation and blood flow alterations [8,9]. This allows for the detection of temperature changes before they can be detected by palpation [10,11] and before the onset of other clinical signs of injury [11,12]. IRT also enables the identification of continuing subclinical changes and allows the verification of the complete clinical healing and, hence, if the horse may return to exercise after required rest [13]. In recent studies, IRT was also applied to interpret changes in the surface temperatures of the thoracic region in the case of back pain diagnosis of equine athletes [14,15], as well as the results of the impact of a load on a saddle [2] or incorrect saddle fit [16]. Moreover, the usefulness of equine IRT in the assessment of transient stress response during training [17,18] and competitive sport [19,20] has been demonstrated. Equine IRT seems to be highly related to thermoregulation, the increase in blood flow due to exercise [18], and the blood concentration of metabolic biochemical measurements [21,22]. During physical exercise, metabolic heat production increases as exercise intensity increases [23], and only a quarter of the energy used by a muscle is converted to mechanical energy. The remaining three quarters are dissipated as heat [24]. Therefore, the radiant energy emitted from the horse’s skin surface may be found as a product of basic metabolic processes, exercise, and pathological conditions. However, it should be kept in mind that the temperature measured from the body surface is related not only to the above internal conditions, but also to the thermal properties of the skin and hair coat and the thermal gradient between the skin surface and the environment [25,26].

It is easy to see that IRT is widespread in the equestrian industry as a valuable tool to monitor the underlying circulation, tissue metabolism, and local blood flow in response to different physiological, pathological, or environmental conditions. However, little or no attention has been paid to the application of IRT in donkeys. The only work the authors are aware of is the study of the effects of season and age on the daily rhythmicity of rectal temperature and body surface temperature during the cold-dry and hot-dry seasons in a tropical savannah [27]. For the infrared measurement, the infrared thermometer and seven landmarks adapted from equine IRT were used. Although this study evaluated differences in the surface temperatures of donkeys of varying age groups under changing environmental conditions, no studies to date have compared the thermal images of donkeys and horses obtained in the same circumstances. The scarcity of works on the imaging of donkeys is a motivation to try to answer the question of whether there are significant differences in the thermal images of horses and donkeys. If the images of these animals were similar, it would suggest that intensively researched methods for analyzing equine images are applicable to donkeys.

In this study, the imaging of horses and donkeys was performed under the same environmental conditions. Following the methodology of previous equine researchers, body surface temperatures in healthy animals were evaluated. The normal thermal image was already described, for e.g., the coronary band [28], distal forelimb joints [26,29], the thoracolumbar region [30], the back, and pelvic regions [31] in the horse. It showed a high degree of symmetry between the left and right sides of the body [1,26] and reproducibility over hourly, daily, and weekly intervals up to 90% [30].

The thermal images were manually segmented into fifteen regions of interest (ROIs) corresponding to underlying large muscles. Since the phenomena observable in thermal images often includes more than one ROI, individual ROIs were combined into groups of ROIs (GORs), and the differences in their mean temperatures were examined. The differences, the occurrence of which was statistically confirmed, constituted thermal patterns, which were the basis for the comparison of both species and the analysis of special cases (outliers). This comparison was the main focus of the experiments in this study. The hypothesis of this study is that the thermal patterns of horses and donkeys are similar.

## 2. Materials and Methods

### 2.1. Animals

Eighteen donkeys (nine mares, seven geldings, and two stallions; mean age 7.78±3.04 years, minimum age 2 years, maximum age 13 years; mean height 119.00±11.72 cm) and sixteen horses (eight mares, six geldings, and two stallions; mean age 7.53±2.83 years, minimum age 2 years, maximum age 11 years; mean height 137.40±9.33 cm) participated in the study. All horses met the growth criteria for ponies, i.e., individual height at withers ≤148.00 cm, according to the standards of the International Federation for Equestrian Sport. However, to facilitate comparison between species, they are called horses throughout the manuscript. Most of the donkeys in the study were mixed breeds; however, the following pedigrees could be listed: two half-breed Romanian donkeys, one half-breed Martina Franca donkey, one half-breed Andalusian donkey, one half-breed Magyar Parlagi Szamér donkey, two quarter-breed Grigio Siciliano donkeys, one quarter-breed Andalusian donkey, five mixed breed donkeys with a quarter-blood Romanian donkey and a quarter-blood Andalusian donkey, two mix-breed donkeys with a quarter-blood Martina Franca donkey and a quarter-blood Andalusian donkey, and two local mix-breed donkeys. Furthermore, one donkey represented a pure bred Poitou donkey (the donkey *D.17* discussed as an outlier case). Horses, in general, represented two typical polish pony breeds (eight Polish Koniks and five Hucul ponies); however, two Haflinger ponies and one half-breed Connemara pony were also included. The donkeys and horses were privately owned and were housed in the same stable located in southern Poland in Lubachów. The owners of the animals consented to our research. The ethics approval was deemed unnecessary according to the regulations of the II Local Ethical Committee on Animal Testing in Warsaw and the National Ethical Committees on Animal Testing because all procedures in the study were non-invasive and did not cause distress and/or pain equal to or greater than a needlestick. The equids were fed three times a day with a dose of hay personalized to each animal to maintain an optimal, healthy condition and had daily access to a grassy paddock no shorter than 8 h per day. All horses received a BCS 3 (body condition score) [32], and all donkeys obtained an FNS 3 (fatty neck score) [33], both on a five-point scale. Both during the study and the month preceding the study, equids were not used in riding, nor were harnessed. Before the IRT imaging, physical examinations were conducted to ensure that the equids were free from preexisting inflammatory conditions. The general examination including the evaluation of internal temperature, heart rate, respiratory rate, mucous membranes, capillary refill time, and lymph nodes and was carried out following international veterinary standards. The detailed examination of the musculoskeletal system was performed following the guidelines for the lameness evaluation of athletic horses [34]. All donkeys and horses were clinically healthy, with no clinical signs of lameness, back problems, or musculoskeletal injury. They had normal species conformation and normal growth pattern. No horses were excluded due to the physical examination results. Two donkeys were excluded due to the properties of their hair coat: the first of them due to hair loss caused by abrasions during transport the week preceding the study and the second due to the significantly longer hair length (7.6±1.2 cm) in comparison with the hair length of the other donkeys (3.4±0.7 cm). Finally, sixteen donkeys were qualified for the formal analysis; however, an analysis of the two donkeys deviating from the accepted uniform appearance is included in the Section 4.2.

### 2.2. Data Collection

To ensure the best possible conditions for the comparison of the collected thermal images, the skin thickness, the hair coat length, and the constant thermal gradient between the skin surface and the environment were taken into account. The study was performed in mid-September, and all measurements were taken on the same day under the same circumstances (ambient temperature 20.2 °C humidity 45%). A total of 68 images were taken in a closed space, protected from wind and sun radiation, to minimize the influence of external environmental conditions [35]. The imaging of donkeys and horses was carried out following equine international veterinary standards [36]. The imaged area was brushed, and dirt and mud were removed 15 min before imaging. The thermal images were acquired on the left and right sides at a 90° camera angle from a distance of approximately 2 m from the animal. During each imaging session, two images of each individual were taken. The images were focused on the center of the trunk. The animals were imaged on the side where the mane was less visible. The images were taken by the same researcher (M.M.) using an infrared radiation camera (FLIR Therma CAM E25, Brazil) with an emissivity (e)∼0.99. The temperature range was standardized in the professional software (FLIR Tools Professional, Brazil) during the preprocessing of the images at the 10–30 °C level.

After each IRT imaging, an ultrasonographic image was taken with an ultrasound scanner (SonoScape S9, SonoScape, Shenzhen, China) using a linear 5–12 MHz transducer (L752, SonoScape, Shenzhen, China). Ultrasound scans were performed with the transducer placed on the animal’s back, over the third lumbar vertebra, perpendicular to the spine. All images were collected on the left side of the animal [37]. The hair was trimmed at the measurement place, and ultrasound gel (Aquasonic 100, Parker Laboratories Inc., Fairfield, NJ, USA) was used as a coupling medium. The real-time ultrasonographic examination was frozen, and the image was saved, as well as the subcutaneous fat (SF) plus skin thickness (SF-Skin) measurements were obtained. An example of an ultrasonographic image is presented in Figure 1. The hair coat samples were taken from the mid-neck approximately 5 cm below the base of the mane. The length of individual hairs was determined from a random sample of five pulled strands, including the roots [38].

#### 2.2.1. Dataset Preparation

Based on collected data, a dataset was prepared that was later used in the experiments. The dataset consisted of images from a thermal camera and the corresponding annotations in the form of class maps of the main muscle areas. Every thermal image was a table of 320×240 pixels. The value in each pixel was the measured temperature value. A corresponding class map was a table, where the value in every pixel was the ROI number, and a value of zero was used for pixels without annotation. An example class map is presented in Figure 2. The class maps were produced by hand annotating the fifteen identified regions of interest (ROIs) in each image. The following ROIs corresponding to the underlying large muscles were annotated:ROI 1 *m. brachiocephalicus*—a parallelogram-shaped area from the lateral surface of the atlas, behind the angle of the mandible, to the regio supraspinatus of the scapula.ROI 2 *mm. splenius capitis and cervicis*—a triangle-shaped area from the lateral surface of the axis to the regio supraspinatus of the scapula above ROI 3.ROI 3 *m. trapezius pars cervicalis*—a triangle ranging from the middle of the neck to the regio cartilaginis of the scapula and along the regio supraspinatus of the scapula up to two-thirds of the length of the scapula.ROI 4 *m. trapezius pars thoracica*—a triangle ranging from the regio cartilaginis of the scapula along the regio supraspinatus of the scapula up to one-thirds of the length of the scapula.ROI 5 *m. latissimus dorsi*—a triangle-shaped area from the regio infraspinatus of the scapula up to two-thirds of the length of the scapula along the back to the tuber coxae.ROI 6 *mm. glutei* (superficialis and medius)—an irregular area in the regio tuberis coxae.ROI 7 *m. biceps femoris*—an oblong s-shaped area in the regio femoris cranially from the m. semitendinosus.ROI 8 *m. semitendinosus*—an oblong s-shaped area in the regio femoris caudally from the m. biceps femoris.ROI 9 *mm. in regio cruris*—a rectangular-shaped area in the regio cruris between articulatio genus and articulatio tarsi.ROI 10 *m. tensor fasciae latae*—an irregular area between the regio tuberis coxae and the flank.ROI 11 *m. obliquus externus abdominis*—a trapezoid-shaped area from the lower two-thirds of the regio infraspinatus of the scapula to the tuber coxae and the regio of processus xiphoideus sterni.ROI 12 *m. pectoralis transversus*—a triangle-shaped area behind the regio olecranon to the regio processus xiphoideus sterni.ROI 13 *mm. in regio antebrachii*—a rectangular-shaped area in the regio antebrachii between articulatio humeri and articulatio cubiti.ROI 14 *m. pectoralis descendens*—an irregular area in the projection of the regio infraspinatus of the scapula.ROI 15 *m. deltoideus*—an irregular area in the projection of the regio supraspinatus of the scapula.

#### 2.2.2. Dataset Availability

In order to facilitate the replication of the experiments presented in this work, the dataset [39] (dataset location: https://zenodo.org/record/4085075) and the experimental source code (source code location: https://github.com/iitis/thermal_patterns.git) are made available to the public under an open license.

### 2.3. Thermal Patterns in the IRT Images of Horses and Donkeys

The main goal of this study was to find patterns in the thermal images of both species and determine if these patterns share similarities. A thermal pattern was defined as a statistically significant difference between the mean temperatures in any two areas composed of groups of ROIs.

#### 2.3.1. Testing the Statistical Significance of Temperature Differences

The statements in this work are usually associated with the comparison of temperatures between areas (subsets of pixels) in a thermal image or images, e.g., a statement “Animals A were warmer than Animals B in area C” means that based on the available sample, the surface temperatures of Animals A were on average higher in this region. Therefore, to test the statistical significance of these statements, the one-sided Mann–Whitney–Wilcoxon (MWW) test [40] was used. MWW is a non-parametric statistical hypothesis test that allows for the comparison of two related sequences of samples. A one-sided test was used because the direction of the difference was known, as it was the average temperature difference in the compared areas. A non-parametric test was used because the temperature distributions in ROIs were diverse and often non-Gaussian. The two sequences of samples were obtained by randomly, uniformly sampling the compared ROIs or groups of ROIs. The number of samples was the size of the smaller set; when comparing individual ROIs between animals, the average difference in the sample count was (10.25±6.58)% of the average ROI size for both species. Unless stated otherwise, the statistical significance of the *p*-value was set at p<0.001.

#### 2.3.2. Finding Thermal Patterns in Animal Species

The definition of thermal patterns in this paper followed the assumption that the surface temperature differences between different parts of the animal’s body repeat within species. To identify these patterns, the following methodology was used:

##### Combining ROIs into Groups of ROIs

In the first step, based on the observation that visible patterns in the thermal images from our dataset were often located in several ROIs, ten groups of ROIs (GORs) were manually designated for analysis. The designated GORs, presented in Figure 3, were as follows:GOR 1 *Neck*, ROIs {1,2,3}—represents an area of skin covering muscles located cranially from the cranial border of the scapula.GOR 2 *Front quarter*, ROIs {1,2,3,4,14,15}—represents an area of skin covering muscles located cranially from the spinous processes of the scapula.GOR 3 *Trunk*, ROIs {5,11}—represents an area of skin covering muscles between the caudal border of the scapula and the vertical line defined by tuber coxae, excluding the area of m. pectoralis transversus.GOR 4 *Hindquarter*, ROIs {6,7,8,9,10}—represents an area of skin covering the examined muscles of the pelvic limbs laid caudally from the vertical line defined by tuber coxae.GOR 5 *Rump*, ROIs {8,9}—represents an area of two ROIs of the pelvic limbs lying the most caudally.GOR 6 *Dorsal aspect*, ROIs {3,4,5,6}—collects an area of skin covering muscles located above the horizontal line halfway up the trunk.GOR 7 *Ventral aspect*, ROIs {9,10,11,12,13}—collects an area of skin covering muscles located below the horizontal line halfway up the trunk.GOR 8 *Abdomen*, ROI {11}—represents an area of skin covering muscles lying between the caudal border of the scapula and the vertical line defined by tuber coxae, excluding the area of m. pectoralis transversus and m. latissimus dorsi.GOR 9 *Groins* (Girth and Flank), ROIs {10,12}—represents two areas of skin mostly covering large muscles of thoracic and pelvic limbs represented by the girth area and flank area.GOR 10 *Legs*, ROIs {9,13}—represents two areas of skin covering muscles of the proximal parts of limbs, both thoracic and pelvic.

##### Comparing GOR Temperatures

In order to compare the average temperatures between the designated groups of ROIs (GORs) and test whether the difference was statistically significant, the following methodology was used:

Let there be set of animals of a given species $A=\{a_{1},\cdots ,a_{16}\}$ and a set of GORs defined in the previous paragraph $G=\{g_{1},\cdots ,g_{10}\}$. Let a set $T_{g}^{a}$ be a set of pixels (temperatures) of an animal a∈A from a group g∈G, and let the mean value of pixels in a set be denoted by δ, e.g., $\delta (T_{g}^{a})$. For every pair of groups (i,j)∈G×G, a difference in average values of temperatures in these groups for all animals was computed, i.e.,
$$\Delta_{\left(i,j\right)}=\delta \left(\underset{k\in A}{⋃}T_{k}^{j}\right)-\delta \left(\underset{k\in A}{⋃}T_{k}^{j}\right).$$

Values $\Delta_{\left(i,j\right)}$ form a matrix of differences $M^{\Delta }\in R^{\left|G|\times |G\right|}$. Due to the fact that values in the matrix $M^{\Delta }$ represent temperature differences, the matrix is not symmetric.

In the next step, the MWW test described in Section 2.3.1 was applied to verify the statistical significance of the difference $\Delta_{\left(i,j\right)}$ for every pair of groups (i,j)∈G×G. This was done in two ways:Globally—for every pair (i,j)∈G×G, the MWW test was applied to the whole population, i.e., the union of sets $\underset{k\in A}{⋃}$
$T_{i}^{k}$ was compared with the union of sets $\underset{k\in A}{⋃}T_{j}^{k}$.Locally—for every pair (i,j)∈G×G, the MWW test was applied separately for every animal a∈A, by comparing the set $T_{i}^{a}$ with the set $T_{j}^{a}$. For the dataset used in this study, this resulted in 16 tests for every pair.

The results of “local” tests formed a matrix $M^{L}\in R_{+}^{\left|G|\times |G\right|}$, where the value in every cell represented the number of animals for which the difference was significant.

##### Thermal Patterns

For a given animal species, a statistically significant difference $\Delta_{\left(i,j\right)}$ between two GORs (i,j)∈G×G was treated as a thermal pattern. A thermal pattern can thus be interpreted as a statement based on available data, e.g., the GOR 5 *Rump* was colder than the GOR 1 *Neck*. If the significance of the difference was confirmed by the “global” test, but not for every animal, by the “local” test, i.e., the value in the matrix $M^{L}$ for this difference was less than 16, and this means that while the pattern emerged in a population, it was susceptible to individual differences of animals; thus, there were animals that did not show this pattern. If the pattern also appeared individually in all of the animals tested, it was considered to be stable.

### 2.4. Thermal Images’ Visualization

In order to visualize the visible structures in IRT images from the dataset used in this study, the temperatures are presented in the form of a color map, modeled on the visible part of the electromagnetic spectrum, i.e., ranging from violet to red. To improve the clarity of the images, the zero values representing the areas outside the ROIs are shown in black. By manipulating the color map threshold values (assigned to its extreme colors), patterns common to all animals or patterns specific to a particular animal are highlighted.

Temperature distributions within a specific ROI are visualized using histograms where the y-axis is presented as a probability density, i.e., bin counts are divided by a total number of counts. Alternatively, boxplots where the box extends from the lower to the upper quartile values are used. The line in the boxplot denotes the median, the whiskers the range of $\left\{q_{1}-1.5*\left(q_{3}-q_{1}\right),q_{3}-1.5*\left(q_{3}-q_{1}\right)\right\}$ where $q_{1}$, $q_{3}$ denote the first and the third quartiles, and circles outliers.

Unless stated otherwise, in all thermal map visualizations, the presented color map temperature values $t_{c}$ are limited to the common range of $t_{c}\in \langle 8.8,30.65\rangle$ °C, which were extreme values in annotated ROIs for all animals included in the study (temperatures in ROIs: horses, $t_{h}\in \langle 10.64,30.65\rangle$ °C, $E(t_{h})=22.72\pm 2.46$ °C, donkeys, $t_{d}\in \langle 8.8,29.56\rangle$ °C, $E(t_{h})=18.88\pm 2.30$ °C), not including the animals *D.17*, *D.18*, for reasons explained in Section 2.1. Alternatively, $t_{c}$ values were selected as extreme temperature values in the ROIs for a given animal to highlight the features of visible thermal patterns; these special cases are clearly indicated.

#### Data Visualization

An individual animal in the dataset could be represented by a vector $v_{i}\in R^{d}$ of *d* features corresponding, e.g., to the means or variances of temperatures in every ROI, which led to d≥15. The extraction and visualization of data structures in a high-dimensional space are often performed by using the principal component analysis [41] (PCA) and projecting data onto the first principal components. However, PCA uses a sample covariance matrix. Since the dataset contained a limited number of examples, the computation of a reliable covariance matrix was difficult. Therefore, the t-distributed stochastic neighbor embedding (t-SNE) [42] algorithm was used for data presentation. T-SNE visualizes data by giving each example a location in a two-dimensional map. An important feature of the t-SNE is that its output is non-deterministic, which results from the fact that the optimization problem solved by the technique has a cost function that is not convex. Since in this work, t-SNE was only used to visualize patterns emerging in the data, it was considered acceptable. The presented visualizations were selected as representative examples after several executions of t-SNE. When t-SNE was applied for data visualization, its perplexity parameter was set to a value of five.

Data features were extracted with common statistics such as the mean, standard deviation, kurtosis, and skewness. In addition, a scenario in which the data were normalized by subtracting the average global temperature of every animal from values of all pixels in this animal’s image was tested.

### 2.5. Implementation

Experiments were implemented in Python 3.6.9 using the libraries: NumPy 1.16.4, SciPy 1.3.1, scikit-learn 0.22.1, Matplotlib 3.2.2, seaborn 0.11.0.

Experiments were conducted using a computer with Intel(R) Core i7-5820K CPU @ 330 GHz with 64 GB of RAM and with the Windows 10 Pro system. The running time of the experiments could be measured in seconds.

## 3. Results

A comparison of temperatures between ROIs is presented in Figure 4. The surface temperatures for horses were, on average, higher than for donkeys, which was confirmed as statistically significant for every ROI (MWW, p<0.001). Considerable variances in ROIs’ temperatures and the presence of many outliers were observed. Example histograms for two ROIs with the most extreme differences in mean temperatures of horses and donkeys are presented in Figure 5. The individual differences in animal surface temperatures resulted in multi-modal temperature distributions, as, e.g., in Figure 5a. Histograms of temperatures for all ROIs can be found in Figure A1 in Appendix A.

A comparison of hair coat length and SF-Skin values between donkeys and horses is presented in Table 1. The skin and the subcutaneous fat were thicker and the hair coat was longer in donkeys than in horses (MWW, p<0.0001).

t-SNE data visualization is presented in Figure 6. When the examples were represented by mean temperature vectors in ROIs, the data formed distinct clusters, separated by low-density areas, as, e.g., in Figure 6a. However, the labels of classes within a cluster were mixed, suggesting that the mean temperature in ROIs alone was not a definite species descriptor. For features based on the standard deviation, clusters ceased to be clearly separable, as, e.g., in Figure 6b. For features based on skewness, kurtosis, or normalized temperatures, it was difficult to observe a consistent clustering of data.

The visualization of thermal maps for horses in this study is presented in Figure 7 and for donkeys in Figure 8. As visual comparison of the images reveals that visible temperature patterns were more complex for horses, e.g.: average temperature values for horses *H.8, H.13* were globally higher; GOR 8 *Abdomen* was visibly warmer for horses *H.4, H.7, H.8, H.3*; and GOR 4 *Hindquarter* was visibly warmer for horses *H.4, H.7, H.8, H.10, H.13*.

The donkey temperatures were more uniform. Temperature values in GOR 5 *Rump* were visibly lower than in other GORs, while in GOR 2 *Front quarter*, warm areas were observed. A comparison of the histograms for four selected GORs is presented in Figure 9, where it can be observed that the overlap between histograms is greater for GOR 2 than for GOR 5.

To highlight the visible patterns, individual thermal maps for two example animals are presented in Figure 10. Visual examination of, e.g., the GOR 2 *Front quarter* in Figure 10c,d reveals how the characteristic patterns are usually associated with groups of ROIs rather than an individual ROI.

Thermal patterns for both species, i.e., the differences in the temperatures between designated GORs, are presented in Figure 11. For both species, GORs *Rump* and *Legs* were consequently colder than the others, while GORs *Neck* and *Front quarter* were warmer. The majority of differences were globally significant (p<0.001). For horses, there were five exceptions: *Neck*/*Front quarter*, *Trunk*/*Ventral aspect*, *Trunk*/*Abdomen*, *Ventral aspect*/*Abdomen*, and *Rump*/*Legs*. For donkeys, there were only two exceptions: *Ventral aspect*/*Abdomen* and *Trunk*/*Groins*.

However, as for the local significance of differences, for horses, there were only five patterns that consequently appeared for all animals: *Dorsal aspect*/*Front quarter*, *Groins*/*Trunk*, *Groins*/*Hindquarter*, *Legs*/*Neck*, *Legs*/*Front quarter*. On the contrary, for donkeys, there were 21 such patterns, which indicates that donkeys were individually more consistent with the global trend.

A summary of the pattern similarities between both species is presented in Figure 12. Figure 12a presents patterns that are similar for both species, e.g., the relation in temperatures between GORs *Rump* and *Neck* was the same for both species (the GOR 5 *Rump* was colder than the GOR 1 *Neck*), and this relation was globally, statistically significant (p<0.001), which is indicated with the green color (the similar and globally statistically significant (SPS) class) in the image. Figure 12b presents the minimal number of animals in each species that shared the corresponding pattern, e.g., for the pair *Rump* and *Neck*, there existed at least 15 horses and 15 donkeys for which the pattern was also locally, statistically significant (p<0.001). The majority of patterns fell under the SPS class, which supports the thesis about the similarities in the patterns for both species. In addition, the dissimilar patterns were most common in GOR 6 *Dorsal aspect* and GOR 3 *Trunk*.

## 4. Discussion

To the best of the authors’ knowledge, this study is the first to describe the whole body surface thermal patterns of donkeys using infrared thermography. Since there are many applications of IRT in horses’ veterinary diagnostic procedures, training monitoring, and welfare evaluations, the initial comparison between species is essential for further donkey IRT applications. Therefore, the motivation of this work is to contribute to a better understanding of the normal thermal pattern for rested donkeys.

The surface temperatures of the horses in our study were on average higher than those of the donkeys, and also, their individual temperatures varied more within the species. This was largely due to the differences in the thermal properties of the skin and the hair coat. The skin and the subcutaneous fat were thicker and the hair coat was longer in donkeys than in horses (see the values in Table 1), providing a better thermal insulation for donkeys. Recent results suggest that the hair coat properties of donkeys and horses are significantly different [38], even in animals with shorter hair than in this study. This difference might be due to the considerably large seasonal variation in hair weight and length typical for horses, but not for donkeys, or different breeds of horses participating in our research (Polish-native warmblooded horses/ponies) compared to [38] (U.K.-native coldblooded horses/ponies).

The use of a heterogeneous group of donkeys and horses in terms of their breed is one of the limitations of this research. It should be taken into account that the horse breed affects coat growth [43]. Therefore, there is a considerable variation in the thermal insulation of the coat between different breeds [44]. In this study, individuals in both groups could differ in skin thickness and/or hair coat, which could influence their thermal insulation and thermal patterns. This is well visible in the case of a long-haired donkey (pure-bred Poitou donkey *D.17*). On the other hand, the SD of each hair length measurement was less than one-quarter of the mean, which indicates an acceptable homogeneity. However, differences in the properties of the hair coat and skin within breeds, as well as other breed-related features could not be excluded as a factor that influenced the presented results. Further studies should consider a larger panel of populations, e.g., different breeds represented by a larger number of individuals. Similar studies were carried out on the warmblooded horses, and the comparison of the superficial body temperatures between thoroughbreds, Arabian, and Polish half-breed horses was reported in [45]. Thoroughbreds were reported as significantly warmer than Arabian and Polish half-breed horses at most ROIs located on a distal part of the limbs and back region. However, no differences in measured temperatures between Arabian and Polish half-breeds were observed [45]. The authors did not find a comparison of whole body temperature patterns between other breeds, especially Polish-native warmblooded ponies, in the available literature. In recent studies, the strong relationship between BCS (body condition score) and SF-Skin, for both donkeys and horses, was demonstrated [37,46]. The higher SF-Skin thickness in donkeys than in horses in this study may indicate greater adiposity of donkeys and thus better insulation. As a result, slight local changes in donkey surface body temperature may be difficult to observe. This makes the warm area visible in regio scapularis associated with the GOR 2 *Front quarter* particularly interesting. Additionally, it suggests the validity of animal temperature analysis through comparing the characteristics of different regions of a given animal.

### 4.1. Similarities in Thermal Patterns of Horses and Donkeys

It was determined in this study that patterns in IRT images were often visible in groups of ROIs, and a methodology of assessing these patterns based on the difference of temperatures in groups was proposed. Figure 12 shows that the thermal patterns for both species share similarities: 77.8% of patterns visible in Figure 12a were similar and statistically significant (p<0.001); 8.9% of patterns were the opposite; and the rest of them could not be statistically confirmed. For 88.8% of globally significant patterns, half or more individual animals from every species shared this pattern. In the authors’ opinion, this supports the thesis about similarities in IRT images of both species.

The results presented in Figure 11 indicate that donkeys were more “uniform” in their GORs, which resulted in larger maximum differences between GORs and the fact that more individual animals shared the global trend, when compared with horses.

The opposite thermal patterns were usually associated with the GORs *Dorsal aspect* and *Trunk*. Both GORs cover a relatively large area of the animal’s body, which raises the question of whether a more granular segmentation of these areas would allow for the discovery of further similarities.

An important question is whether the trends observed in the studied dataset are characteristic of the entire population. As the number of cases was limited by practical considerations, we believe that our results should be treated as a significant indication of the existence of the relationships described. At the same time, we emphasize the need to further verify these conclusions in a larger population. To facilitate this, all research data related to this study are available to the public under open licenses. However, the authors speculate that many imaging approaches applied successfully in equine veterinary medicine [1,25] can also be used for donkey IRT imaging, subject to the visualization conditions presented here. The IRT may become a useful diagnosis tool in donkeys, particularly in the field of orthopedics [5,6,7,14] or effort assessment [21,22,24], but even in the field of animal welfare [2,18,33]. Given the increase in general interest in donkeys, as milk producers, companion animals [33], and working animals under a saddle and in a harness [47], the new noninvasive imaging approaches in donkeys are in demand.

### 4.2. Special Cases

The two animals identified as outliers, i.e., *D.17–18*, allowed for an interesting study of how general or specific the proposed thermal patterns were. The visualization of the differences between the patterns of these animals and the rest of the donkeys is presented in Figure 13. In Panels b and c, the green color indicates the compliance of the animal thermal pattern with the global trend, i.e., the difference for a given pair of GORs has the same sign, and it is statistically significant (p<0.001) for the animal. Thermal patterns for the donkey *D.17* with a thick hair coat were usually in line with the global pattern, except for four pairs of GORs, where the statistical significance of the local animal pattern could not be confirmed. The patterns of the donkey *D.18* were much less in line with the global pattern. This was an expected result, as the patchy hair losses visibly affected its thermal characteristics in the image. This also suggests that the proposed thermal patterns may be the basis for creating temperature indexes, as, e.g., in [2], or features for detecting anomalies. The temperature difference matrices for these two animals are provided in Figure A2 in Appendix A.

## 5. Conclusions

The results of this study indicate that the characteristic thermal patterns of both horses and donkeys are mostly associated with groups of ROIs (GORs) rather than an individual ROI. Based on this observation, the thermal pattern is defined as a statistically significant difference between the mean temperatures of the designated GORs for a given animal species. The thorough verification of the significance (both globally for all data and locally for individual animals) reveals the similarity for the majority of proposed thermal patterns in both studied species. It is worth noting that the thermal patterns of the donkeys are more uniform than those of the horses, and the donkeys are individually more consistent with the global trend. The average surface temperatures compared within the proposed thermal patterns are higher for the studied horses than for the donkeys, which may be related to different thermal properties of their skin and hair coat.

## Figures and Tables

**Figure 1 animals-10-02201-f001:**
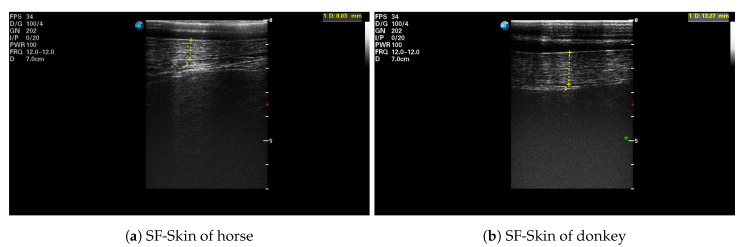
Example of an ultrasonographic image taken over the third lumbar vertebra: (**a**) the horse *H.1*; (**b**) the donkey *D.3*. The subcutaneous fat plus skin thickness (SF-Skin) is highlighted.

**Figure 2 animals-10-02201-f002:**
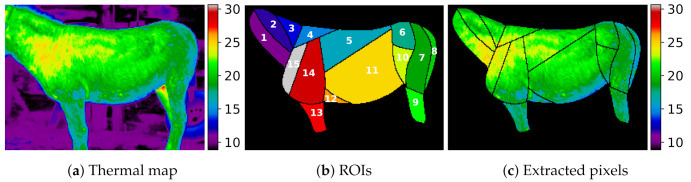
Visualization of a donkey *D.3*: (**a**) thermal data from the camera as a thermal map; (**b**) annotated classes corresponding to selected ROIs (see Section 2.2.1); (**c**) extracted pixels used in the experiments

**Figure 3 animals-10-02201-f003:**
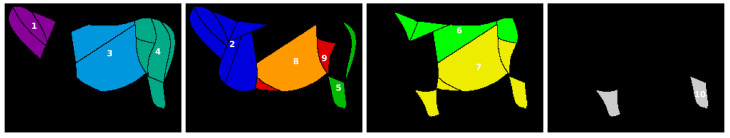
Visualization of a donkey *D.3*, divided into groups of ROIs (GORs).

**Figure 4 animals-10-02201-f004:**
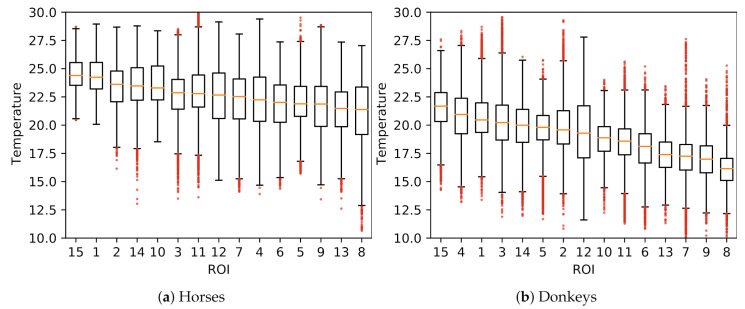
Visualization of temperatures in ROIs (ordered by their medians) of all animals: (**a**) horses; (**b**) donkeys.

**Figure 5 animals-10-02201-f005:**
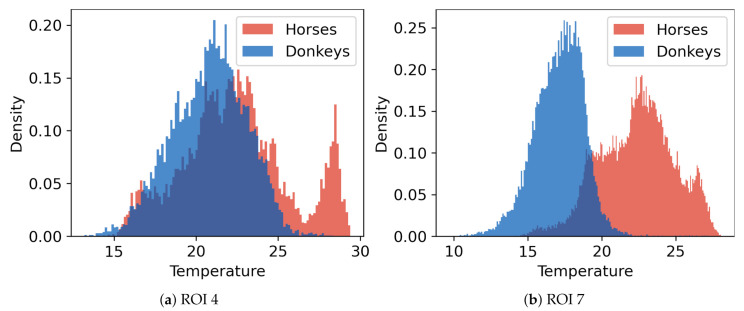
Histograms of temperatures for two ROIs where the difference ∆*_t_* between mean values of temperatures for the two animal species is: (**a**) the smallest (ROI 4, ∆*_t_* = 1.59) and (**b**) the largest (ROI 7, ∆*_t_* = 5.26).

**Figure 6 animals-10-02201-f006:**
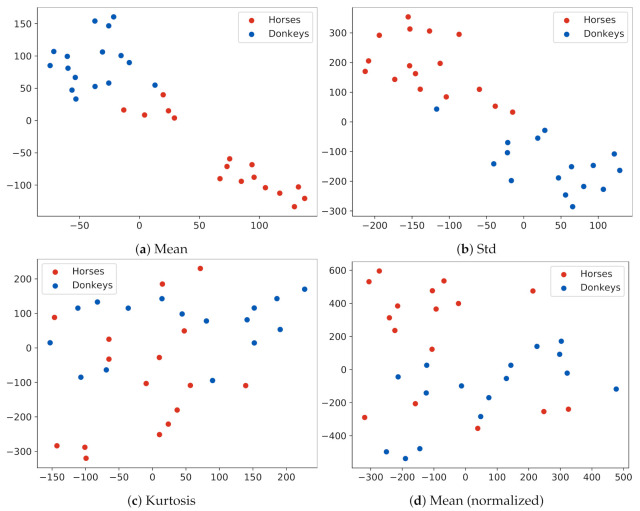
. t-SNE visualization of the dataset. Every dot represents an animal described with features extracted from the pixels of its 15 ROIs. Plots present different feature extraction statistics: (**a**) the mean; (**b**) the standard deviation; (**c**) the kurtosis; (**d**) the mean, after removing the global mean temperature of an animal from all pixel values. Notice that the examples in Plot (**a**) form clusters that correspond to the species of the animal, although some examples are in the wrong cluster.

**Figure 7 animals-10-02201-f007:**
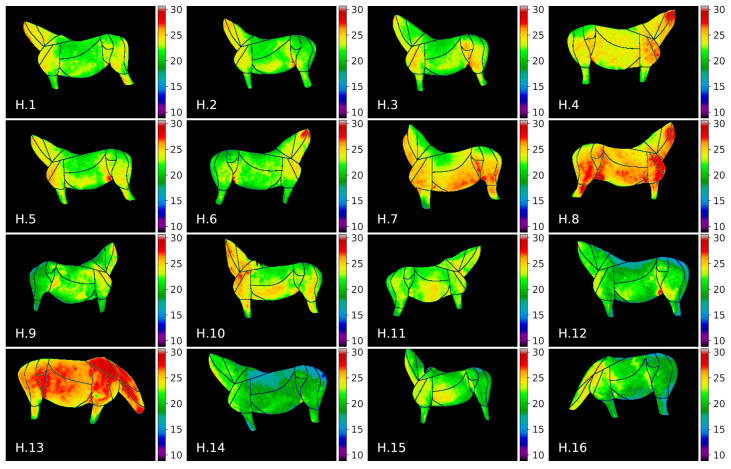
Thermal maps of annotated ROIs for horses in our dataset.

**Figure 8 animals-10-02201-f008:**
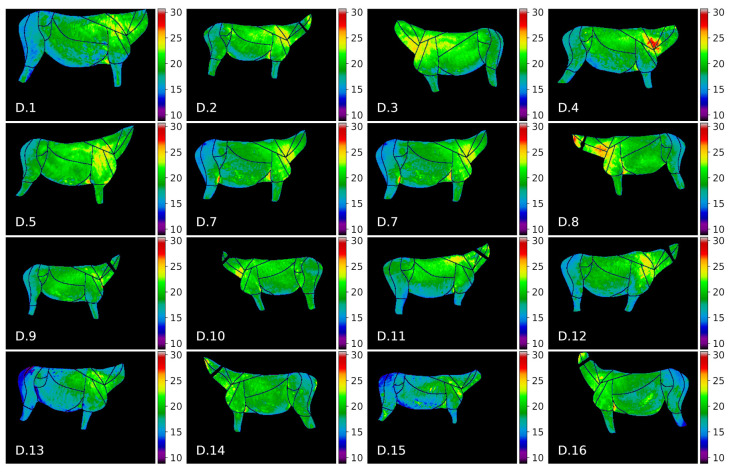
Thermal maps of annotated ROIs for donkeys in our dataset.

**Figure 9 animals-10-02201-f009:**
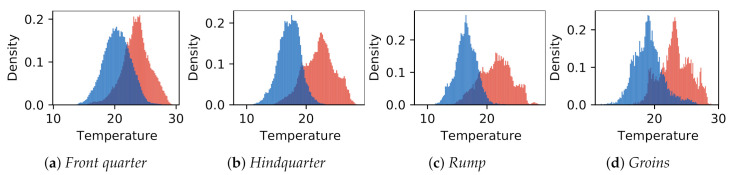
Comparison of temperature histograms between animal species in identified characteristic areas corresponding to selected groups of ROIs: (**a,**) GOR 2 *Front quarter*; (**b**) GOR 4 *Hindquarter*; (**c**) GOR 5 *Rump*; (**d**) GOR 9 *Groins*. Horses are represented in red and donkeys in blue.

**Figure 10 animals-10-02201-f010:**
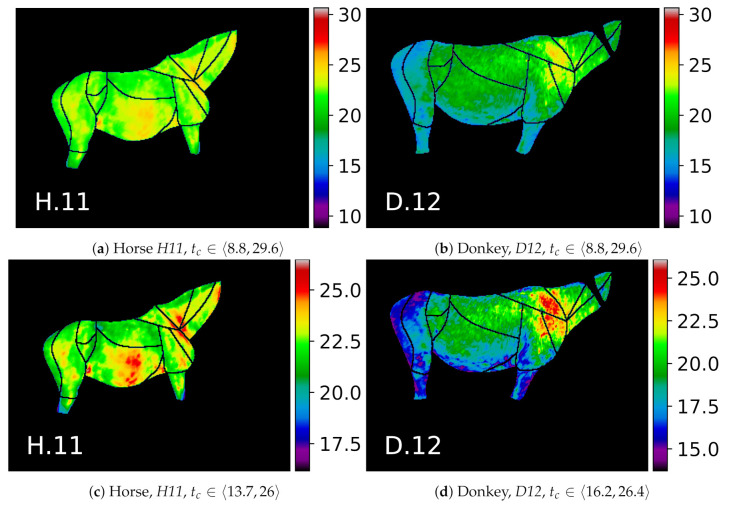
Selected examples of two animals from the dataset. The color map values *t_c_* for images in the upper row are scaled to the common range, which makes them easy to compare: (**a**) horses; (**b**) donkeys. Images in the bottom row are scaled to the minimal and maximal temperatures in the annotated ROIs of each animal, which highlights individual thermal patterns: (**c**) horses; (**d**) donkeys; e.g., warm horse’s GORs *Abdomen* and *Neck*, cool donkey’s GOR *Rump*, and warm donkey’s GOR *Front quarter*.

**Figure 11 animals-10-02201-f011:**
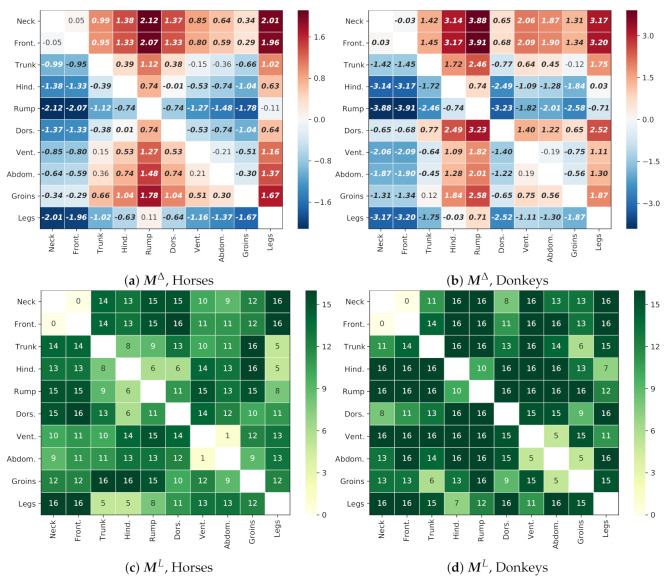
Thermal patterns and statistically significant differences between GORs. The upper panels present the matrix of differences within one species: (**a**) horses; (**b**) donkeys; e.g., the value $M_{\left[4,0\right]}^{\Delta }$ = −2.12 in the cell [4, 0] in Panel (**a**) is the difference between the mean temperatures for the pair *Rump* and *Neck*, indicating that the *Rump* GOR is colder. Bold font indicates “global” statistical significance of this difference (*p* < 0.001). The bottom panels present tables for (**c**) horses and (**d**) donkeys, with the number of animals for which the corresponding temperature difference in the table above is statistically significant considering individual thermal pattern of this animal (*p* < 0.001); e.g., the value $M_{\left[4,0\right]}^{L}$ = 15 in Panel (**c**), which indicates that the pattern *Rump* and *Neck* is locally significant for 15 horses. A stable pattern should be statistically significant simultaneously for all data combined and for each of the 16 animals of a given species.

**Figure 12 animals-10-02201-f012:**
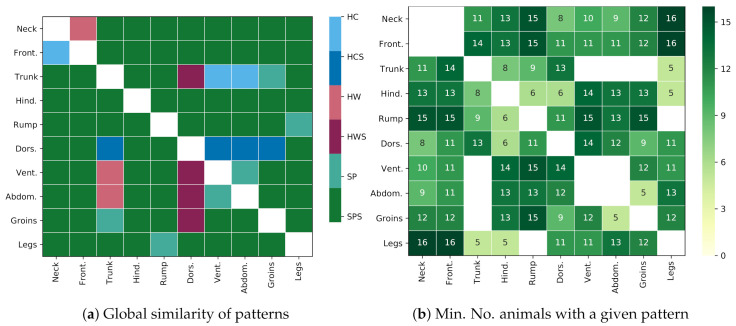
Comparison of thermal patterns for both species: (**a**) division of thermal patterns into six classes: SPS denotes thermal patterns that are similar and globally statistically significant (*p* < 0.001) for both species; SP: similar patterns, but not significant; HWS: opposite patterns where horses are warmer (and donkeys colder), which are statistically significant; HW: same as HWS, but not significant; HCS: significant patterns where horses are colder (and donkeys warmer); HC: same as HCS, but not significant. Note that the SPS class is the most common, which suggests the global similarity of patterns. (**b**) The minimum number of animals that locally confirm the global trend for classes SPS, HWS, and HCS, i.e., for both species, at least this number of animals share a given pattern individually (*p* < 0.001). Note that the maximum value in the table is 16, which indicates a stable pattern.

**Figure 13 animals-10-02201-f013:**
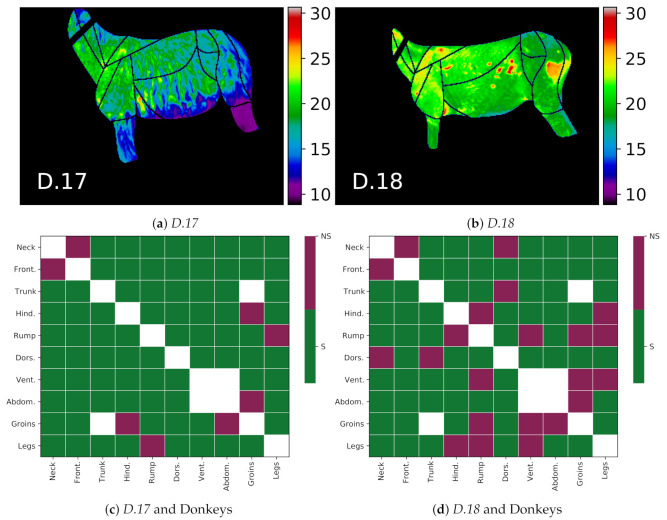
Differences between donkeys *D.17* and *D.18*, outlier cases, and the rest of the donkeys, i.e., animals *D.1-16*. The upper panels present thermal maps of the two cases: (**a**) donkey *D.17*; (**b**) donkey *D.18*. Donkey *D.17* was colder than other animals due to its long hair length. Donkey *D.18* had an unusual pattern of warm areas resulting from patchy hair loss. Bottom plots show differences in the thermal patterns of these donkeys compared to the global pattern of other donkeys: (**c**) donkey *D.17* compared to other donkeys; (**d**) donkey *D.18* compared to other donkeys. The *S* class (green) indicates that the individual animal pattern was in line with the global trend, and class *NS* (red) indicates the opposite.

**Table 1 animals-10-02201-t001:** Measured features (mean ± SD) of horses (*H.1*–*H.16*) and donkeys (*D.1*–*D.16*): the length of the hair coat and the thickness of the subcutaneous fat plus skin (SF-Skin).

Animals	Hair Coat (cm)	SF-Skin (mm)
Donkeys	3.39±0.46 *^a^*	12.01±0.83 *^c^*
Horses	1.78±0.38 *^b^*	8.80±0.87 *^d^*
*p*-value	<0.0001	<0.0001

Different superscript letters indicate significant differences between horses and donkeys for hair coat (*a*, *b*) and SF-Skin (*c*, *d*) respectively according to the Mann–Whitney–Wilcoxon (MWW) test.

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
