# Peer review of "Comparison of the Surface Thermal Patterns of Horses and Donkeys in Infrared Thermography Images"

_animals, 2020, doi:10.3390/ani10122201_

Round 1

Reviewer 1 Report

The reserach aim to verify if the IRT used for horses is usefull also for donkeys, and could be of interest for the readers. However, M&M paragraph is difficult to read and understand due to the great quantity of information and level of details. I suggest to semplify and move some information in a supplementary material section, in order to make the paper more readible. 

In the same way, it is not very clear to the reader how the results could be applied on the donkey. I suggest to improve this argumentation, in order to better connect your results with practical aspects that moved the aim of your research. 

It is not clear if you used only one breed of horses or more. I think this should be mentioned, as a different breed might influence the results for the different skin or hair coat. Please mention this also in the discussion as a limit of the study.

Please replace in the text the present form with the past form of the verbs.

Other comments in the attached file.

Author Response

Thank you very much for your comments and a substantial amount of time invested in looking over the manuscript. We are very pleased with your opinion, that our work could be of interest for the readers. We have a sincere hope that after thorough improvement following your comments our manuscript will be worth to be presented for the wider community.

We have addressed all of the comments and modified the paper accordingly.  Our detailed answers follow. Please note that reviewers’ comments are in italics while our answers are not.

Reviewer 2 Report

Dear authors,

Please see the recommendations, corrections and questions formulated during the revision process of your manuscript and use as considered to increase the value of your paper.

Simple Summary

L3: Please spell out all abbreviations at first usage

L4: Do you refer literally rested or being at rest (i.e. resting)? Consider to change if it’s the latter

Abstract

Please use impersonal voice, here and throughout the manuscript (for own work)

Introduction

L27: insert ‘it’ before ‘is’

L37: You may wish to include a sentence on the ability of IRT to identify continuing subclinical changes e.g. clinically symptomless ‘healed’ injuries (i.e. not only showing the changes before becoming clinically evident but to confirm/infirm clinical healing as a measure to schedule returning to exercise after an injury that needs resting)

L74: Please use past tense, here and throughout the manuscript, where you describe the work you DID (including the description of setting)

Materials and Methods

L77: ‘This section describes the methodology’ is a redundant phrasing as the title already states this. Please rephrase

L90: ‘without obesity’ is redundant, as optimal, healthy condition exclude obesity

L93: Replace ‘were’ to ‘was’

L95: Change ‘of’ to ‘to’

L98: whereas

L109: Change ‘minute’ to ‘minutes’

L112: Insert ‘The’ before ‘Images were…’

L115: insert ‘the’ before ‘images at…’

Table 1: The table doesn’t present just the average hair length and skin thickness for horses and donkeys but the statistical significance of the calculated differences between species, which is brilliant, but these are already results which belong to the Results section…

L171: ‘Here we describe our methodology’ – please remove/rephrase as it is redundant

L183: Please rephrase ‘usually’, it doesn’t sound scientifical

L183: Please change ‘require’ (The statistical significance of p was set at…). Also take care of the PAST tense of the work performed IN THE PAST in this section too (and throughout the manuscript)

L186: ‘The general idea is’ doesn’t fit well in a scientific paper, please rephrase

L187: Insert ‘than’ after ‘surface temperature’

L188, 189: Please change ‘dependencies’, it is not the appropriate word

L195-L215: What do you mean by the ‘impact of muscles’? Please find a better phrasing for the readers’ understanding. Honestly, I don’t get what you meant.

L218-219: Please use impersonal voice! Please use only past tense when you refer to what you did for your study. Remove ‘We do it as follows’ (and other similar throughout the paper)! Please consider something as: In order to compare the average temperatures between the designated groups of ROIs and to test the statistical difference the following formula was used: ………. Where: and list all the meanings of symbols etc

L220: Rephrase, please. ‘The obtained differences were presented as a matrix (formula) which was not symmetric as showed…..’ or similar

Please use the recommendation for the whole section!

L228-231: Please don’t state here what (and how) you will present in the Results section

L238-239: Please be more exact in describing your criteria to consider a pattern stable and reliable (the terms ‘most’ and ‘more’ may be insufficient for repeatability of the study)

L243: Please don’t say what you describe, just do it! Consider to rephrase to ‘thermal and data visualization techniques were used to illustrate the observations of the study’ or similar

L 269-273: Please don’t use future tense to describe past actions

Results

L276: Remove this sentence (However, the ‘Results’ section should not describe experiments)

L 277-290: These paragraphs doesn’t belong to this section, please move them

L278: Remove the comma after ‘otherwise’

L279: Please rephrase the sentence with the appropriate usage of ‘either’

Figure 4 title: Consider to remove ‘ROIs were’

L294: Rephrase excluding ‘we can see’, here and later on too

L297: Consider to rephrase to ‘Temperature distributions can be multi-modal, resulting from…’ or ‘ The individual differences in animal surface temperatures resulted in multi-modal temperature differences’ or similar

L299-300: This was already said in the ‘Materials and methods’ section, where it belongs, please rephrase

L302-303: Please clarify. What do you mean by ‘wrong’ cluster? Were the animals misclassified or not? ‘could be’ is misleading here

L306: Please be clearer about the meaning of ‘usually’

L310: Consider to change to “As the visual…’ and replace the colon with a comma

L313: Insert ‘to be’ after ‘seem’ and replace the comma with a dot after ‘uniform’. Remove ‘we notice’. Use past tense

L315: Remove ‘we can see’ here and in the followings

L321: Remove ‘prepared using the methodology described in Sec. 2.3.2’

L329: Remove ‘which will be further addressed in the discussion’

L333-334: State statistical significance threshold in brackets, here and elsewhere as needed

L338: Remove ‘we notice’

L339-340: Remove ‘The possible explanation of this observation will be discussed in the next section.’

Figure 6 title: Rephrase by: removing ‘i.e.’ and ‘(see Sec. 2.3.3.)’; changing ‘genre’ to ‘species’; stating the value of p for all statistical significances found; considering to move the exemplifications out of the title

Figure 7: State p values

Discussion

L342: Remove the commas before and after ‘on average’

L344: Rephrase to ‘The skin and the subcutaneous fat were thicker and the hair coat was longer in donkeys than in horses (see values in Tab 1), providing a better thermal insulation for donkeys.’ Higher thickness and hair length sound odd

L346: Change ‘results’ to ‘research’

L347-348: Change ‘differed significantly’ to ‘are significantly different’, don’t end the sentence here and continue by replacing ‘Although the authors indicate, contrary to us, a lower hair length in donkeys than in horses’ with ‘even in animals with shorter hair than in our study’ or something similar. There is no contradiction between the two studies, just a difference. Insert ‘difference’ between ‘This’ and ‘may’, replace ‘may’ with ‘might’

L352: Change ‘were’ to ‘was’

L353: Change ‘isolation’ to ‘insulation’

L361-362: Remove ‘Visualisation in’, ‘looking at’ and use impersonal voice

L363: Remove ‘We also see’

Figure 12 title: Replace ‘are’ with ‘in’

L366: Rephrase excluding ‘we can see’

L369: Rephrase excluding ‘we notice’, replace ‘associates’ to ‘associated’

L371: Delete ‘of’ and change ‘will’ to ‘should’

L372: Remove the comma after ‘question is’, change ‘for our data set’ to ‘in the studied data sets’

L382: Insert ‘it’ before ‘is’, state statistical significance value in brackets

Conclusions

Please rephrase using impersonal voice and simpler formulation. A rewrite for guidance would be as follows:

As the results of the study showed, the characteristic thermal patterns of both horses and donkeys were mostly associated with groups of ROIs (GORs) rather than an individual ROI. Based on this observation the thermal pattern was defined as a statistically significant difference between designated GORs for a given animal species. The thorough verification of the significance (both globally for all data, and locally for individual animals) revealed similarity for the majority of proposed thermal patterns in both the studied species. Noteworthy the thermal patterns of donkeys were more uniform then those of the horses, and donkeys were individually more consistent with the global trend. The average surface temperatures compared within the proposed thermal patterns were higher for the studied horses than for donkeys, which may be related to different thermal properties of their skin and hair coat.

Figure 13 title: Remove ‘Visualisation of’, ‘which were identified as’ and ‘(See Sec. 2.1.)’. Remove “Notice that’. Use past tense.

Figure A2: move from the Reference list

Title: Remove ‘(see. Sec. 2.1.)’. State the value of p for statistical significance

L430: Correct the referenced title (‘Digital’ instead of ‘Igital’. The title is listed incorrectly on the publication’s website, indeed, but once opened the article has the correct title, as you could have seen)

Author Response

Dear Reviewer, we sincerely thank you for your response and the valuable comments on our manuscript. We are truly grateful for your remarks which have allowed us to improve the manuscript considerably. We sincerely hope that after all these improvements our work will be worth to be presented to the wider community.

We have addressed all of the comments and modified the paper accordingly.  Our detailed answers follow. Please note that reviewers’ comments are in italics while our answers are not.

Author Response

Thank you very much for your kind words about our work, and your opinion that our work provides an advance towards the current knowledge and is relevant for both the clinical and welfare elds. We have given these experiments, methods, data, and software a lot of work, and we hope that it is clear, well systematized, suciently informative and easy to replicate. We are truly grateful for your comments which have allowed us to improve the manuscript considerably. We hope that we have achieved adequate quality of the manuscript by incorporating the suggested changes.

We have addressed all of the comments and modified the paper accordingly.  Our detailed answers follow. Please note that reviewers’ comments are in italics while our answers are not.

Round 2

Reviewer 1 Report

The suggestions have been properly addressed.